# Excitonic signatures of ferroelectric order in parallel-stacked MoS$_2$

Swarup Deb [1] ✉, Johannes Krause[1], Paulo E. Faria Junior [2], Michael Andreas Kempf[1], Rico Schwartz [1], Kenji Watanabe [3], Takashi Taniguchi [4], Jaroslav Fabian[2] & Tobias Korn [1] ✉

Interfacial ferroelectricity, prevalent in various parallel-stacked layered materials, allows switching of out-of-plane ferroelectric order by in-plane sliding of adjacent layers. Its resilience against doping potentially enables next-generation storage and logic devices. However, studies have been limited to indirect sensing or visualization of ferroelectricity. For transition metal dichalcogenides, there is little knowledge about the influence of ferroelectric order on their intrinsic valley and excitonic properties. Here, we report direct probing of ferroelectricity in few-layer 3R-MoS$_2$ using reflectance contrast spectroscopy. Contrary to a simple electrostatic perception, layer-hybridized excitons with out-of-plane electric dipole moment remain decoupled from ferroelectric ordering, while intralayer excitons with in-plane dipole orientation are sensitive to it. Ab initio calculations identify stacking-specific interlayer hybridization leading to this asymmetric response. Exploiting this sensitivity, we demonstrate optical readout and control of multi-state polarization with hysteretic switching in a field-effect device. Time-resolved Kerr ellipticity reveals direct correspondence between spin-valley dynamics and stacking order.

The basic building block for any interfacial ferroelectrics is two layers of certain van der Waals materials aligned parallel to each other[1–11]. In the case of transition metal dichalcogenides (TMDs) like MoS$_2$, layer-asymmetric atomic registry[12] along the out-of-plane direction leads to interlayer hybridization between the valence band of one layer and the conduction band of the other but not vice versa[13]. Such a preferential coupling induces unidirectional charge transfer, leading to the spontaneous emergence of an electric dipole bound at the interface. A bilayer unit of parallelly-stacked (rhombohedral or so-called 3R)-TMDs thus possesses two equivalent yet opposite ferroelectric orders viz. MX and XM, marking the stacking of metal (M) and chalcogen (X) atoms at the eclipsed sites (Fig. 1).

Besides hosting ferroelectricity, parallel stacking of TMDs preserves the spin-valley locking, regarded predominantly as a property of monolayers[14], even in the multilayer limit due to broken inversion and mirror symmetries[15–18]. Therefore, the 3R-polymorph of TMDs provide means for the bottom-up construction of three-dimensional spin-valleytronic devices on a ferroelectric platform. A visionary goal would be to engineer the optical response in TMDs by exploiting the ferroelectricity-induced interaction, which can be highly localized, non-volatile, and reconfigurable. Despite a tremendous interest in this emerging phenomenon[19,20], so far the field has been limited to indirect sensing or visualizing the ferroelectricity employing surface-sensitive probes, such as atomic force microscopy[2,4,6,7,11], scanning electron microscopy[21], or sensing-layer-based approaches[2,5] and, therefore, could only provide limited physical insight. In the case of TMDs, this leaves a void of in-depth knowledge about the influence of ferro-electric order on their intrinsic valley and excitonic properties. To fill this gap, we exploit hyperspectral reflectivity imaging to discern the correspondence between ferroelectric stacking and optical response

[1]Institute of Physics, University of Rostock, Albert-Einstein-Str. 23, Rostock 18059, Germany. [2]Institute for Theoretical Physics, University of Regensburg, 93040 Regensburg, Germany. [3]Research Center for Electronic and Optical Materials, NIMS, 1-1 Namiki, Tsukuba 305-0044, Japan. [4]Research Center for Materials Nanoarchitectonics, NIMS, 1-1 Namiki, Tsukuba 305-0044, Japan. ✉e-mail: swarupdeb2580@gmail.com; tobias.korn@uni-rostock.de

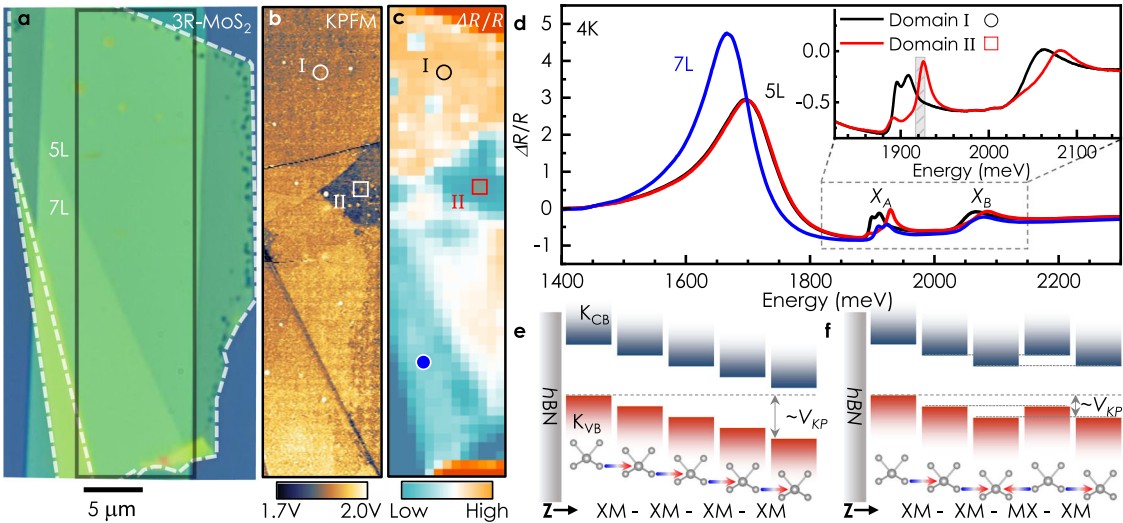

**Fig. 1 | Reflection contrast imaging of ferroelectric domains in few-layer 3R-MoS₂. a** Optical micrograph of a 3R-MoS₂ flake on Si/SiO₂/hBN. Topographical steps and edges of the bottom hBN have been marked by white dotted lines. **b** Surface potential map within the area enclosed by the black rectangle in (**a**). Two domains, marked by ○ and □, can be identified by the difference in contrast. **c** Integrated intensity map of $\Delta R/R$ at the $X_A$ spectral region, i.e., from 1906 to 1918 meV (gray bar in inset of **d**). **d** Low-temperature reflectance contrast spectra from various spatial locations marked by symbols of corresponding color in c

(black - 5L domain I, red - 5L domain II, and blue - 7L). Inset shows high-resolution spectra collected from domain I and II. **e, f** Two crystal configurations of 5L 3R-MoS₂ overlayed on layer-projected $K_{VB}$ and $K_{CB}$ band-edges. To first order, the band edge variation along Z and the degeneracies, highlighted for the XM-XM-MX-XM stacking by dotted lines, result from the ferroelectricity-induced electrostatic considerations (SI Figs. S12, 13 for other configurations). Arrows represent polarization vectors at the interfaces.

in these structures. We demonstrate ferroelectricity-induced control of spin-valley dynamics by systematically studying the ultrafast response of excitons using transient reflection and Kerr ellipticity. As-grown samples are chosen over artificially stacked parallel structures, which provides a moiré-free crystal, enabling us to explore excitonic phenomena of purely ferroelectric origin and to employ far-field optical spectroscopy on mesoscopic domains with well-defined stacking. Clear signatures of stacking-dependent K-valley excitonic response and a striking contrast in valley relaxation dynamics are observed. This presents the prospect of using spin and valley as useful degrees of freedom in a robust multilayer platform. It is worth mentioning that recent experimental efforts[22] also underscore the possibility of utilizing multilayer 2H-TMDs for valleytronic applications despite their centro-symmetric structure[23].

## Results

### Ferroelectric domains in 3R single crystals

We begin by probing the electric surface potential of deterministically stamped 3R-MoS₂ flakes on non-polar hexagonal boron nitride (hBN) exfoliated on Si/SiO₂(-285 nm) substrate (SI. S1). An atomic force microscope operated in side-band Kelvin probe mode (KPFM) (SI. S2) is used to map the surface potential, $V_{KP}$. Figure 1a shows an optical microscopy image of a representative flake (Sample 1) composed of five and seven layers of MoS₂. In a multilayer flake, each of the interfaces may have either XM (polarization pointing upward) or MX (downward) stacking order creating a potential ladder either going upwards or downwards, respectively[7,13]. The difference in the total number of upward- and downward-pointing polarized interfaces decides the cumulative polarization state, hence the surface potential. Therefore, different ferroelectric domains in a given thickness would be visible in a $V_{KP}$ map. Based on this understanding, the existence of two ferroelectric domains, viz. domain I and II in the 5L region of the sample, becomes evident from Fig. 1b. The measured surface potential variation of ～127 mV between these domains (SI Fig. S1) corresponds

to the sum of effectively two ferroelectric interfaces, consistent with earlier results[7,24].

The appearance of different domains in a single crystal flake can be attributed either to inherent stacking faults in the bulk 3R crystal or to the sliding by slip-avalanche of constituting layers from XM to MX stacking order or vice versa. Since the energies of all possible stacking configurations are similar (SI Fig. S19), the latter occurs as a result of inadvertent shear strain during exfoliation[25] or deterministic stamping. To further illustrate the mechanism, we fabricated a sample by intentionally applying an in-plane perturbation, achieved by laterally tapping the substrate holder during the dry transfer of 3R-MoS₂ from PDMS to the substrate. This horizontal force introduces a shear stress between the layers that remain anchored to PDMS and the layers attached to the substrate. This deliberate horizontal force, which is significantly stronger than the ubiquitously present shear forces during standard operations, generates smaller and densely packed ferroelectric domains, as identified by KPFM (SI Fig. S2). The ferroelectric domains, in this case, are of similar dimension as in the artificially stacked moiré interfacial ferroelectrics. However, achieving precise control over domain formation and their structural organization remains a challenge.

### Correspondence of surface potential and exciton fine structure

To check for the ferroelectric-field-induced changes in optical response, we map the exciton transition energies over the entire flake (i.e. Sample 1) using confocal differential reflectance spectroscopy (SI. S3). Typical reflectance contrast (RC) spectra, $\Delta R/R = (R_{Ref} - R_{Flake})/R_{Flake}$[26] obtained at different spatial locations of the flake at liquid helium temperature are presented in Fig. 1d. Here $R_{Flake}$ is the reflection from MoS₂ and $R_{Ref}$ is the same from hBN (SI Fig. S3). In the energy range of interest, the reflectance contrast of the few-layer MoS₂ is dominated by three excitonic features, viz. the momentum-indirect transition at ～1700 meV and the well-known A and B excitons at $X_A$ ～1900 and $X_B$ ～2050 meV[27–29]. Notably, the lowest energy feature

despite being a momentum-indirect transition has a prominent appearance. This can be attributed to the favorable thin-film interference condition in the Fabry-Pérot cavity formed by Si/SiO$_2$/$h$BN/MoS$_2$ in the given energy range (SI Figs. S4, S5 for additional examples and further discussion). We note that RC spectra, $\Delta R/R$ can also be calculated using $(R_{Ref} - R_{Flake})/R_{Ref}$[30]. We found that both representations, $\Delta R/R_{Ref}$ and $\Delta R/R_{Flake}$ share similar characteristic viz. the presence of three distinct features and their energetic positions (SI Figs. S4, S5, S6, S15). However, the choice of definition modifies the line shape of each transition and therefore, the overall appearance of the spectra. Evidently, the lowest energy peak appears vividly in $\Delta R/R_{Flake}$ (SI Figs. S4–S5, S10). Therefore, in the following we refer to $\Delta R/R_{Flake}$ as RC spectra unless mentioned otherwise.

We proceed by constructing RC maps corresponding to different transitions by integrating the intensity over selected spectral ranges. The map of the momentum-indirect transition brings out a clear contrast between the five- and seven-layer regions (SI Figs. S7–S8) due to the red-shift in its spectral position with increasing thickness. On the other hand, the $X_A$ intensity map exhibits signatures of two distinct domains within the topographically smooth 5L area, similar to the KPFM map. The contrast in the intensity map (Fig. 1c) is due to the appearance of higher- and lower-energy sub-features in the spectral region from 1880 to 1930 meV. The energy positions and relative oscillator strengths of these fine features distinguishably mark the ferroelectric stacking configuration, as revealed by the high-resolution spectra shown in Fig. 1d-inset. In contrast to previous studies on artificial R-stacked TMDs[31], these subfeatures are not related to nanoscale reconstruction of moiré domains (as confirmed by the complementary KPFM measurements). A similar spatial map is obtained from room temperature spectroscopy measurements (SI Fig. S9); however, due to thermal broadening, spectral features can not be fully resolved. Correspondence between RC maps and KPFM images has been observed in several samples of different thicknesses, see Fig. S10 for example. Upon closer inspection, it becomes evident that $X_B$ exhibits a similar trait (SI Figs. S7, S8). To improve the signal-to-noise ratio of the RC spectra around the $X_B$ transition, we exploit the cavity effect by carefully choosing the bottom $h$BN of our stack. SI Fig. S11 shows an RC spectrum of a trilayer-MoS$_2$ flake on a 95 nm-$h$BN/285 nm-SiO$_2$/Si substrate, where the splitting in the $X_B$ becomes apparent. In a nutshell, momentum-direct excitons at the K points in the Brillouin zone reveal the stacking order, while the spectral feature related to the momentum-indirect transition remains insensitive to it. Based on the trend of red-shift[32] with increasing layer number, as shown in Fig. S4 and previous studied on similar systems[29,33], one can tentatively assign the low-energy peak either to the $\Gamma$-Q or to K-Q hybrid-excitonic[34] transition.

To explain the asymmetric response of different exciton species to ferroelectric ordering, we consider the band structure of 3R-MoS$_2$. It has been shown that in bilayer 3R-TMDs, states at the $\Gamma$ point of the valence band (VB) or Q point of conduction band (CB) are fully delocalized due to strong interlayer hybridization[7,13,16]. In contrast, the wavefunctions at the K points are almost entirely layer-localized. The ferroelectricity-induced built-in potential manifests as a local electrostatic perturbation and introduces a shift between the layer-projected energy levels at K points, leading to a type-II band alignment at the interfaces[16], with the energies decreasing from layer X to layer M. By the same token, in a multilayer sample, the spatial profile of K-point band extrema along the out-of-plane direction follows the underlying ferroelectric potential and is, therefore, susceptible to the stacking order, as depicted schematically in Fig. 1e–f and SI Figs. S12, S13.

Neglecting interlayer coupling through hybridization, a built-in electric field should shift the conduction and valence bands by the same amount, leaving the transition energies unchanged. Clearly, this alone is insufficient for interpreting the experimentally observed emerging multiplicity of transition energies. Therefore, to this end, we

inspect the stacking-dependent strength of K states delocalization based on our band structure calculations depicted in Fig. 2a–c. Figure 2d–f, g–i presents the five lowest energy conduction-band and highest-energy valence-band spin-up electron wavefunctions at the K point, sorted according to their energy eigenvalues (for the following sections, we have dropped the term 'K point' while referring to bands and wavefunctions unless stated otherwise). For the fully polarized, XM-XM-XM-XM case (Fig. 2a, d, g), the individual wave functions both at the valence and conduction band remain layer-localized with their energy levels corresponding to the ferroelectric potential ascending with each additional layer. However, for other variants, interlayer hybridization is evident (e.g., Fig. 2b, e, h, c, f, i). This dependence of hybridization on ferroelectric ordering is the underlying mechanism for the observed contrast in optical spectra. Before proceeding further, we note that a five-layer 3R-MoS$_2$ crystal can have sixteen ($2^{N-1}$, $N$ = 5) stacking configurations, out of which KPFM can distinguish among five combinations based on their surface potential (SI Fig. S12). At this stage, therefore, instead of attempting a one-to-one quantitative description of the stacking-dependent excitonic response, we focus on the qualitative generic trend with three specific cases as examples (all the studied systems are shown in the SI Figs. S20–S27).

In the fully co-polarized XM-XM-XM-XM stacking, the potential varies monotonically from layer to layer[7]. Thus, the three central layers experience a similar ferroelectric field - preserving the energy degeneracy of their intralayer excitonic transitions, appearing as a single vertical line in the dipole matrix element plot (Fig. 2j). However, the top and bottom layers show energetically distinct transitions due to broken translational symmetry at the surfaces. The scenario changes in partially polarized stacking due to selective interlayer coupling between specific layers. The mechanism described below is similar to the quantum correction that leads to level anti-crossing by lifting the energy degeneracy of coupled quantum systems. For instance, in the XM-XM-MX-XM configuration, electrostatic degeneracy (cf. Fig. 1f) leads to layer-specific resonant hybridization[35] of $K_{VB}$ states, which is particularly strong between the second and fourth layers. Pronounced hybridization also occurs in the XM-XM-MX-MX case between the mirror-symmetric layers, namely the first and fifth, as well as the second and fourth. In contrast to the valence band states, the $K_{CB}$ states largely retain their layer-localized nature in almost all cases except mirror-symmetric stacking orders. This layer- and band-specific strength of interlayer hybridization leads to unequal shifts of layer-projected conduction and valence bands, lifting the intralayer transition degeneracy even for the center layers and increasing the maximum spread of energies, clearly visible in Fig. 2j–l. These findings are of a general nature for any parallel-stacked multilayer TMDs. This stacking-specific resonant hybridization is not relevant for the $\Gamma$ and Q states, which have a significant degree of hybridization irrespective of stacking.

In addition to the corrections to the electronic band structure arising from hybridization, changes in the exciton binding energy due to asymmetric dielectric surroundings also play a role in determining the energetic position of individual transitions[36]. In order to provide a qualitative comparison to the experimental reflectance contrast spectra, we calculate the absorption spectra based on the intralayer dipole matrix elements and exciton binding energy (SI. S5, S6), considering a phenomenological Lorentzian broadening with a full-width-half-maximum of 20 meV. These results are depicted in Fig. 2m. Evidently, the general shape with multiple sub-features, which strongly depend on the individual stacking, closely mimics the experimental observations. However, it is important to note that DFT typically underestimates the band gap of semiconductors due to the unaccounted derivative discontinuity of the typical exchange-correlation functionals[37] and, therefore, the calculated absorption spectra are redshifted by about 480 meV in comparison with the experimental results. By applying a rigid energy shift to the band gap, estimated from GW

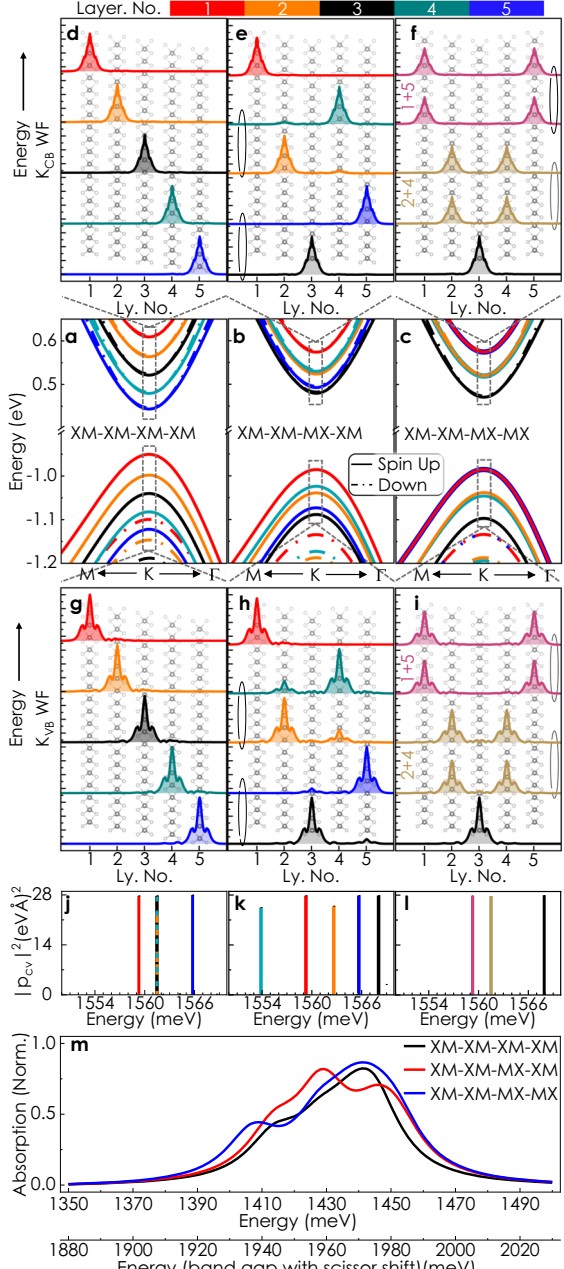

**Fig. 2 | Ab initio calculation of stacking-dependent band structure, wave-functions and optical response. a–c** Band structure for different stacking orders. Spin orientation (∥ $Z$) of the individual bands is indicated by solid (spin up) and dashed (spin down) lines, respectively. The bands are colored corresponding to the dominant contribution of specific layers. The stacking of the band extrema at K follows the underlying ferroelectric potential. Due to mirror symmetry for the XM-XM-MX-MX case, some bands are (nearly) degenerate and cannot be attributed to an individual layer (indicated by different colors in **f, i**). **d–i** Absolute value of the lowest-energy spin-up conduction band (**d–f**) and highest-energy valence band (**g–i**) wavefunctions sorted according to their energies superimposed over the crystal structure. Colors indicate the dominant layer(s) for each wavefunction. Ellipses are used to indicate (near-)degenerate states. The wave functions are layer-localized for the fully polarized case (**d, g**). For XM-XM-MX-MX (**f, i**), the lowest-energy wave function is localized in the center layer, while the other wave functions have equal weight in the mirror-symmetric layers. **j–l** Oscillator strength for transitions from all spin-up $K_{V B}$ to all $K_{CB}$ with $s^+$ polarization. The magnitude of these intralayer transitions is in good agreement with values for pristine monolayer $MoS_2$[56,57]. **m** Calculated absorption spectra in the A excitonic region based on intralayer dipole matrix elements and layer-specific exciton binding energy, including a phenomenological Lorentzian line width of 20 meV.

calculations, one can achieve transition energies that are more consistent with experimental data. For $MoS_2$, this estimate yields a band gap increase of about 530 meV[38], which nearly matches the experimentally observed energies.

These observations provide an alternative experimental approach to identify local changes in ferroelectric ordering in 3R-TMD flakes without resorting to surface-sensitive[5–7,39] measurements, which become a challenge for heterostructures. In contrast to other optical measurement schemes relying on a sensing layer[40], the TMD itself reveals its ferroelectric order in optical spectra. This allows for identifying local domains, even in 'buried' layers, as required for integrating more complex, functional heterostructures based on ferroelectric TMDs.

## Non-volatile electrical control and optical readout of multi-state polarization

Our findings, therefore, present an unprecedented opportunity of exploiting ferroelectric fields to gain non-volatile control over excitonic transitions in a field effect transistor architecture- a crucial step forward for applications. We prepare devices on a conducting $Si/SiO_2$(-90 nm) substrate, which acts as the gate electrode, and use few-layer graphite flakes to make electrical contact with $3R-MoS_2$ (viz. Sample 4). We use fully $h$BN-encapsulated structures for these measurements (therefore, $R_{Flake}$ is the reflection from $h$BN/$MoS_2$/$h$BN heterostructure and $R_{Ref}$ is the same from $h$BN/$h$BN). Here, we present results obtained on a trilayer sample at room temperature. First, we map the ferroelectric domains at a few different gate voltages. Two different domains are visible in the false color map as blue and orange regions. Application of positive gate voltage results in an expansion of the blue domains at the cost of the orange domains. An opposite change is observed for negative gate voltage. Next, we measure the reflection contrast close to a domain boundary as a function of gate voltage. The $X_A$ and $X_B$ transitions becomes weaker as the gate voltage is swept forward from -18 V to 18 V, corresponding to increasing electron doping. This reduction is due to a combined effect of Pauli blocking and reduced oscillator strength with state filling - typical of n-type TMDs[41–43]. Notably, the reduction of intensity from the highest to lowest occurs through two distinct steps during the sweeping process (Fig. 3e). Furthermore, a prominent hysteretic behavior in the intensity of $X_A$ and $X_B$ transitions is observed as the gate voltage decreases during a backward sweep, for which the intensity rises again and saturates (Fig. 3b–e and SI Fig. S16). We also examine the standard deviation, which correlates with the full-width half maximum of $X_A$ and $X_B$. Evidently, the resulting plots exhibit the same features (SI Fig. S17).

The step-like features indicate discrete changes in optical response due to ferroelectric switching through a mechanism known as slidetronics, where the out-of-plane electric field induces an in-plane sliding of adjacent layers[4,21,44–46]. At high electric fields, all the interfaces align to achieve an energetically favorable configuration, viz. MX-MX or XM-XM, as indicated in Fig. 3e. Therefore, the hysteretic occurrence of the discrete switching events in response to the external field directly results from a finite coercive field - a hallmark of ferroelectric materials. The two distinctive steps observed for the trilayer sample reveal that switching between the multiple ferroelectric states of a multilayer is feasible and controllable via external voltages - a pivotal insight for scalability.

Our study yields an average coercive field of $0.03-0.035$ $Vnm^{-1}$ for domain wall motion. Moreover, the hysteretic behavior of the monotonous intensity variation with gate voltage indicates a stacking-dependent change in average carrier concentration, observed earlier in transport measurements[6,47]. We note that a complete ferroelectric switching is not achieved throughout the whole crystal within the experimentally accessible gate voltage. This can be attributed to the increasing domain boundary deformation energy[48] and domain wall pinning at localized stacking faults as well as surface contaminants[21].

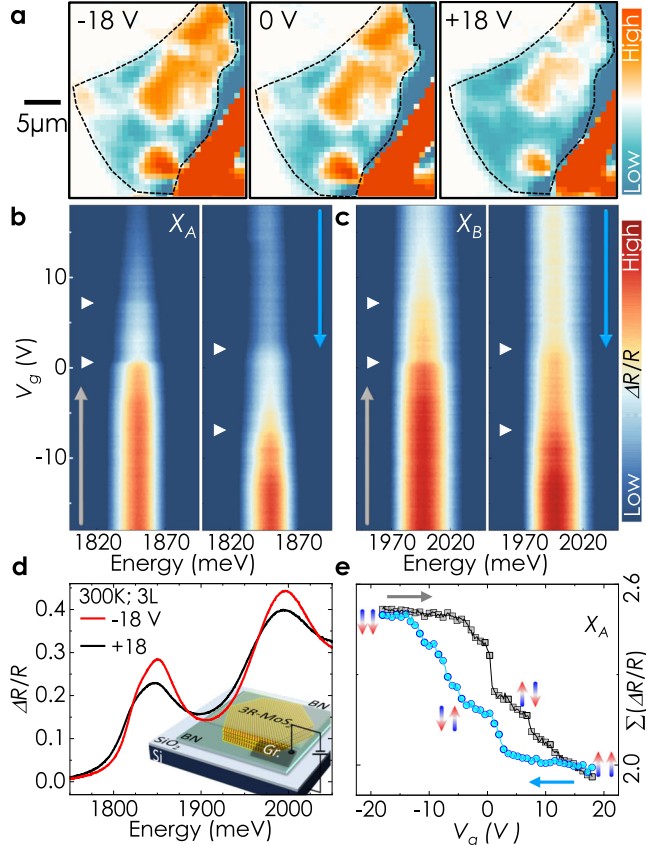

**Fig. 3 | Hysteretic reflectance contrast as a function of gate voltage. a** Integrated amplitude false color map of the numerically calculated first derivative of RC spectra at different gate voltages to visualize domain switching. All measurements were performed at room temperature. Evidently, the out-of-plane electric field drives the blue-colored region to grow in area coverage with increasing gate voltage at the expense of the orange-colored region. **b, c** False color map of A and B exciton intensity as a function of gate voltage during forward and backward sweep. They share the same y-axis, given on the left. **d** Two representative room temperature reflectance contrast spectra at opposite gate voltages. Inset- schematic illustration of the field effect device. **e** Integrated intensity profile ( ≡ vertical line cuts from **b**) at $1852 \pm 2$ meV. The vertical arrows represent the ferroelectric ordering of each interface.

## Domain-resolved exciton population and spin-valley dynamics

Besides gaining control over the excitonic oscillator strength, a meticulous understanding of excitonic lifetime is essential in the pursuit of realizing exciton-based circuits. To this end, we focus on probing the stacking-order-dominated exciton dynamics and spin relaxation using two-color pump-probe Kerr microscopy. Figure 4a illustrates the measurement scheme. We use a circularly polarized pump pulse, slightly blue detuned to $X_A$, to induce a spin polarization of chosen helicity. The photo-generated excitons lead to photo-bleaching of a given valley, resulting in helicity-dependent reflectivity. Therefore, the transient exciton population and its spin-valley polarization can be probed by recording the intensity and induced ellipticity of a time-delayed linearly polarized probe pulse after reflection[49,50]. A series of simultaneous transient reflectivity (TRΔR) and transient ellipticity (TRKE) measurements are performed in which the probe laser wavelength is tuned across the energy range of $X_A$ with the pump energy set at 1958 meV. Figure 4b, c are false-color representations of TRΔR obtained from domain I and II of Sample 1 (cf. Fig. 1), respectively, as a function of pump-probe delay. The energy-dependent temporal evolution of reflectivity in individual domains clearly corresponds to their

respective steady-state response (superimposed white lines). For both the domains, TRΔR traces at the higher resonance (marked by blue arrows) evolve similarly viz. an ultrafast decay component followed by relatively slower dynamics, as depicted by the blue colored traces in Fig. 4d, e. The response at the lower resonance energy has different characters for different domains. For instance, in domain I, we observe a monotonous decrease in the signal from t=0. Remarkably, for domain II, during the initial five ps, the reflectivity of the probe pulse increases with time, followed by a slower decay. The initial increase likely stems from slow state filling at the probe energy during relaxation, possibly suggesting interlayer charge transfer prompted by interlayer hybridization. These observations, therefore, further substantiate the theoretical understanding developed in the context of steady-state optical response.

Unlike the transient reflection, the ellipticity signal manifests no prominent energy dependence. Instead, it is remarkably sensitive to the ferroelectric stacking configuration and exhibits entirely different traits for the two domains investigated. Figure 4f and g portrays the TRKE results for domain I and II probed at higher and lower resonance energies (SI Fig. S28 for the complete spectrum). In domain I, the TRKE decay is relatively slow, albeit faster than the decay of the transient reflectivity traces, indicating that it is not predominantly driven by exciton recombination but by a spin dephasing mechanism on a similar timescale as observed in $MoS_2$ monolayers[51,52]. By contrast, in domain II, the excitons lose their spin polarization almost instantly. The microscopic mechanism for this striking dependence of exciton lifetimes and spin-valley dynamics on ferroelectric order remains an open question. However, recent calculations[53,54] show that in parallel-stacked TMDs, dependent on the symmetry of the stacking order, in-plane spin-orbit coupling terms can arise that lead to spin mixing of K states. In this way, specific stacking orders may facilitate spin dephasing, while others protect spin orientation, yielding prolonged spin-valley lifetimes comparable to TMD monolayers. This opens up the use of spin and valley as useful degrees of freedom in 3R-stacked multilayers, which have so far been limited predominantly to monolayer TMDs.

To conclude, we demonstrate the direct correspondence between ferroelectric order and intralayer excitons in 3R-stacked $MoS_2$ multilayers, allowing us to map ferroelectric domains directly using far-field optical spectroscopy. Ab initio theory provides a comprehensive explanation of the microscopic mechanism. In a field-effect device, we are able to control and map domain wall motion at room temperature and observe clear hysteretic features with a coercive field of roughly 0.03–0.035 V nm$^{-1}$. The relatively small switching field of ferroelectricity by domain wall sliding could facilitate efficient electrical control of various optical properties of 2D materials[55]. Addressing individual domains in time-resolved experiments, we find that depending on ferroelectric order, 3R-stacked $MoS_2$ multilayers can show spin-valley dynamics on timescales comparable to TMD monolayers or near-instantaneous loss of valley polarization. Our study paves the way for utilizing ferroelectric order in few-layer TMDs embedded within functional van der Waals heterostructures as a local, nonvolatile control lever for excitonic and valleytronic properties. Theses findings introduce a distinctive perspective to the field of quantum optoelectronics with van der Waals materials and will enable novel optoelectronic device architectures that rely on efficient electrical manipulation of excitons or the optical response in general.

## Methods
### Device fabrication

3R-MoS$_2$, obtained from HQ Graphene, were exfoliated onto polydimethylsiloxane (PDMS). MoS$_2$ flakes were selected according to their optical contrast. Chosen TMD flakes were stamped on pre-exfoliated $h$BN flakes. We use a polycarbonate (PC)/PDMS-based hot pickup

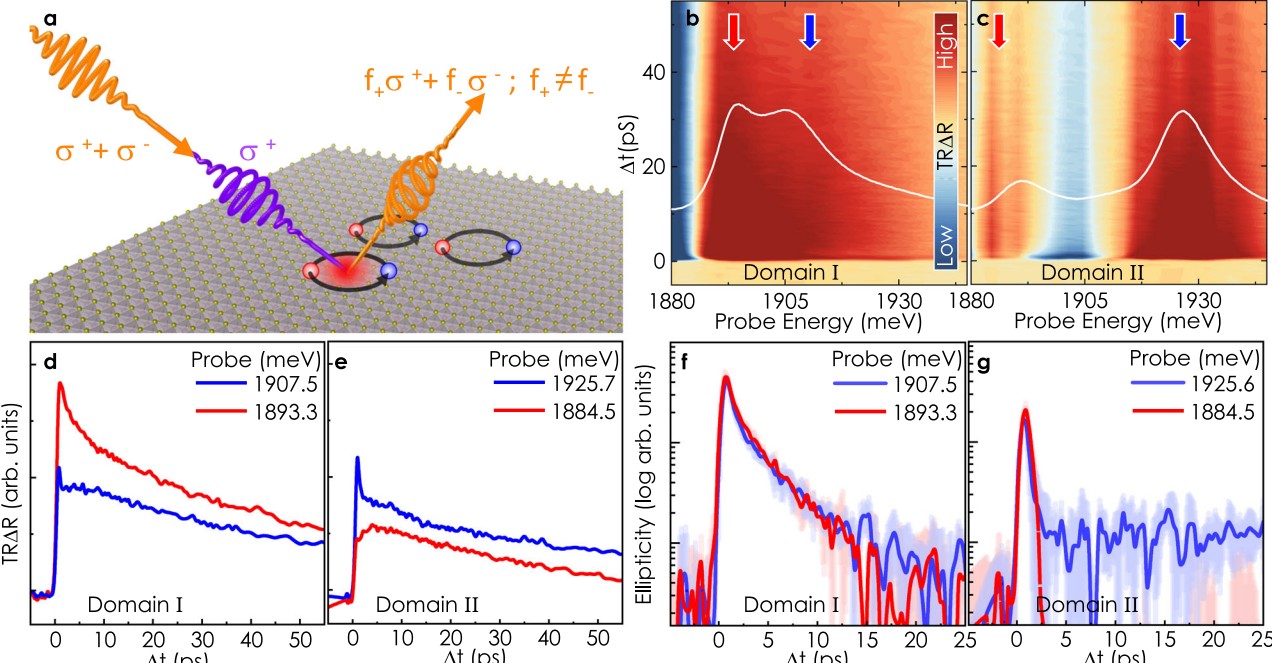

**Fig. 4 | Exciton population and spin dynamics. a** Schematic of pump-probe Kerr measurement. A blue detuned (to $X_A$) circularly polarized light pulse for the pump and linearly polarized pulse for the probe were used. Upon reflection, the linearly polarized pulses turn elliptically polarized. By projecting the reflected elliptic pulse on a quarter waveplate and a Wollaston prism, we can separate out the relative strength of right and left circular components. The total probe intensity change is equivalent to net exciton density, and the intensity difference between left and right circular components gives the net spin-valley polarization. All the pump-probe measurements were done at 4 K. **b, c** False color map of transient reflection as a function of probe energy from domain I and II. White contours (without y-scale) are copied from Fig. 1d-inset for comparison. **d, e** Transient reflection at selected energies, marked by colored arrows in (**b, c**). The opposite trend in the amplitude of the TRΔR signal at the lower and higher energy states from the two domains is a direct consequence of opposite relative strength of these sub-features in the steady-state spectroscopic signal. **f, g** Traces of transient ellipticity at resonant energies in respective domains. The spin-valley lifetime in domain I is on the order of 10-15 ps, while in domain II the signal decays down to the detection noise-floor on sub-ps timescales.

method to transfer hBN flakes from bare Si/SiO₂ substrate to encapsulate hBN/TMD stack. A few layer thick graphite flakes (procured from NGS Naturgraphit), exfoliated on PDMS, are deterministically stamped to make external contact to the sample.

## KPFM measurements

KPFM measurements were performed using Park System NX20. The electrostatic signal was measured at side-band frequencies using two built-in lock-in amplifiers. We used PointProbe Plus Electrostatic Force Microscopy (PPP-EFM) n-doped tips. For the unencapsulated sample (Sample 1 c.f. Fig. 1) we used non-contact mode of operation. The average height above the surface was controlled via a two-pass measurement. The first pass records the topography, whereas, in the second pass, the tip follows the same scan line with a predefined lift (typically 4–5 nm) and measures the KPFM signal. In case of Sample 4 c.f. Fig. 3 where we used a ~ 85 nm hBN for top encapsulation, we operated the AFM in tapping mode and the KPFM measurements were done in a single pass.

## Reflectance measurements and Spatial filtering

For reflectance measurements, the samples were illuminated with a quartz tungsten halogen lamp. The collimated beam was focused on the sample using an 80X objective (NA=0.50, f=200). The reflected light from the sample was collected using the same objective and spectrally resolved using a spectrometer and a charge-coupled detector. In the detection path, before the spectrograph, we introduce a home-built *Spatial-Filtering* module to enhance the spatial resolution. The input side of the spatial filter consists of an aspheric lens mounted on a Z-translator. It focuses the reflected beam onto a pinhole. The pinhole was carefully aligned to the beam path with the help of an XY translator. A fraction of the focused light passes through the pinhole aperture. Another aspheric lens was used to collect the beam from the pinhole and collimate along the detection path.

## Time resolved reflectivity and Kerr ellipticity

In the pump-probe setup, two separately tunable pulsed lasers (Toptica: femtoFiberPro) were used. Each system emitted with a pulse repetition rate of 80 MHz, a spectral width of 6 meV, a pulse duration of about ~ 200 fs. Probe and pump pulses were electronically time-synchronized with aa auto- and cross-correlation width of ~ 300 and 700 fs, respectively. The cross-correlation time of ~ 700 fs marks temporal resolution of our measurement. The pump beam was circularly polarized through an achromatic λ/4 plate. The pump beam was set to a power of 30 μW and the probe beam was set to 20 μW. Both the beams were focused with an 80X (NA=0.50, f=200) objective onto the sample. Therefore, the pump and probe fluence roughly correspond to a maximum of 1.7 and 1.1 KW/cm², respectively. After reflection the pump beam was filtered out by a long-pass filter. The polarization of the probe beam was analyzed for its ellipticity by a combination of a λ/4 plate, a Wollaston prism, and two photodiodes (ThorLabs PDB210A Si Photodetector). The difference signal of the two diodes was fed into a lock-in amplifier, yielding a TRKE signal. The sum signal of the diodes was fed into a second lock-in amplifier, yielding the TRΔR signal.

## Data availability

The data generated in this study have been deposited in the FigShare database under accession code https://figshare.com/projects/ExcitonicSignaturesOfFerroelectricOrderInParallel-stackedMoS2/217963.

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

## Acknowledgements

The authors would like to thank I. Barke for technical assistance. S.D. acknowledges financial support by the Humboldt foundation and a startup funding grant provided by the DFG via SPP2244. The authors also acknowledge financial support by the DFG via the following grants: SFB1277 (project No. 314695032), SFB1477 (project No. 441234705), SPP2244 projects FA791/8-1 and KO3512/6-1 (project No. 443361515), INST 264/181-1 FUGG (project No. 441219355) and KO3612/7-1 (project No. 467549803). K.W. and T.T. acknowledge support from the JSPS KAKENHI (grant numbers 21H05233 and 23H02052) and World Premier International Research Center Initiative (WPI), MEXT, Japan.

## Author contributions

S.D. and T.K. conceived the idea for the study. S.D. performed the experiments, prepared the samples together with J.K., analyzed the data together with T.K. and P.F.J. and wrote the manuscript in close interaction with all other authors. M.K., R.S., and T.K. designed, built, and tested the setup for the time-resolved Kerr experiments. S.D. upgraded the setup for simultaneous time-resolved reflectivity measurements. K.W. and T.T. provided high-quality *h*BN crystals for sample preparation. P.F.J. and J.F. performed the DFT calculations and analyzed the results together with S.D. and T.K.

## Funding

## Competing interests

The authors declare no competing interests.
