## [Peer Review File · Nature Communications]

Excitonic signatures of ferroelectric order in parallel-stacked MoS₂REVIEWER COMMENTS

Reviewer #1 (Remarks to the Author):

This article entitled "Excitonic signatures of ferroelectric order in parallel-stacked MoS₂." proposes optical diagnostics for interfacial ferroelectricity. The observations are supported by good-quality experiments and theoretical studies. The authors find the memory effect in a moire-free 3R-MoS₂ crystal by the proposed method. The results of the manuscript are potentially interesting to related fields. Nevertheless, the main experiment needs to be described more carefully. The following points should be clarified or modified before publication:

1. Authors should carefully describe how they calculated $\Delta R/R$. The intensity or reflectance spectroscopy for R_{ref} or R_{flake} could be included in the supplementary information.
2. What is the definition of R_{ref} ? Is it the position of hBN/SiO₂/Si without MoS₂ for Fig. 1c? What is the blue circle in Fig. 1c? Where is R_{ref} for the hBN-encapsulated MoS₂ in Fig. 3b?
3. The authors defined the contrast spectrum according to the definition in reference [20] and related it to the absorption coefficient. However, I wonder if this definition is correct. $\Delta R/R_{ref}$ seems to be correct according to past literature (Surface Science, 24 (1971) 417). If the definition is wrong, all results need to be recalculated.
4. As the authors explain, $\Delta R/R$ is a physical quantity associated with the absorption, which is related to the thickness of the hBN film. Quantitative labels for the intensity of the reflection contrast should include the axis in Fig. 1d and Fig. 3, as in Fig. S5 or Ref [20] for the validity of experimental results.
5. The energy is of XB at 2000meV in Fig. 3 smaller than in Fig. 1. What is the reason for the difference?

Reviewer #2 (Remarks to the Author):

Due to the broken inversion symmetry resulting from a unique atomic registry, bilayer 2D materials such as hexagonal boron nitride and transition metal dichalcogenides (TMDs) can exhibit interfacial ferroelectricity, leading to rich physics within this fascinating system. While TMDs display distinct excitonic properties with wide-ranging optoelectronic applications, the impact of interfacial ferroelectricity on these properties remains largely unexplored. In this study, focusing on five-layer 3R-MoS₂, the authors identified two types of ferroelectric orders within a single MoS₂ flake and demonstrated the significant influence of these orders on intralayer excitonic transitions. Leveraging this finding, the oscillation strength of intralayer excitonic transitions was utilized as a non-invasive indicator to track the evolution of ferroelectricity orders under an external electric field. Additionally, the study revealed substantial differences in exciton population and spin-valley dynamics under the two ferroelectricity order conditions, with the authors proposing several explanations for these observations.

While the investigation topic holds promise, the manuscript currently lacks robust evidence

to fully substantiate its arguments. Consequently, in my opinion it may not meet the standards required for publication in Nature Communications. Below are some questions for reference.

Q1: Fig. 1d, peak assignment of the Γ -Q transition might be suspicious. As claimed in the Abstract, “momentum-indirect excitons with out-of-plane electric dipole moment” refers to the Γ -Q exciton. Momentum-indirect transitions mostly manifest small oscillation strength in the optical reflectance/absorption spectrum since this physical process needs the assistance of phonons. Meanwhile, according to the manuscript, the optical measurement was conducted in a back-scattering geometry, while the Γ -Q exciton dipole moment is out-of-plane orientated, which translates to that the incidence parallels to the dipole moment, and this would further suppress the optical response of the Γ -Q transition in the reflectance spectrum. However, Fig.1d shows a much pronounced peak than the momentum-direct exciton transitions. For the author’s reference, the intense peak at lower energies were not observed in similar systems. (Nat. Nanotechnol. 2023, 18, 36–41; PRB. 2022, 106, 045411), and the authors would better provide more convincing evidences.

Q2: The authors claimed that the two-domain structure was probably introduced by the shear strain during exfoliation, which causes sliding of constituent layers. In addition to the RC sub-features, the authors are encouraged to provide more evidence to exclude the moiré-ferroelectricity scenario.

Q3: Fig. 3e, there is an evident discontinuity in the junction of the forward and backward sweeping curves at $\sim \pm 0.18$ V nm⁻¹, what could be the reason leading to this phenomenon?

Q4: Fig. 4d and 4e, the slow decaying of lower energy level signal in domain II was attributed to the slow state filling resulted from the interlayer charge transfer probably facilitated by the interlayer hybridization, however, comparing the amplitude of the TR Δ R signal of the lower and higher energy states, why the two domains exhibit an opposite behavior? Also, some experimental details are missing, such as pump fluence, instrument response function.

Minor typos, Supporting Information, on page 1: “hot pickup method [cite] to transfer hBN flakes...”; “...a conductive coating acquired from XXX”.

Reviewer #3 (Remarks to the Author):

In the manuscript “Excitonic signatures of ferroelectric order in parallel-stacked MoS₂”, the authors use the reflectance contrast spectroscopy to reveal the ferroelectricity in few-layer 3R-MoS₂. They show that contrary to electrostatic perception approaches, the hyperspectral reflectivity imaging is able to discern the correspondence between ferroelectric stacking and intralayer excitons. They attribute the origin of this correspondence to the stacking-specific interlayer resonant hybridization by means of ab initio calculations. They realize the room temperature control and optical readout of multi-state polarization with hysteretic features in a field-effect device. They also reveal the spin-valley dynamics on timescales comparable to TMD monolayers. Overall, the manuscript is well prepared and may be valuable for multiple fields of ferroelectrics, optoelectronics, and valleytronics. I can recommend its publication if the following comments are addressed.

1. The asymmetric response of the momentum-direct excitons at the K point is caused by the interlayer hybridization, but the essential relationship between ferroelectric ordering and interlayer hybridization is not clear. I understand that the ferroelectric ordering would lead to a built-in electric field which shift the conduction and valence bands, but it does not change the transition energy and thus has no contributions on the asymmetric response, as mentioned in the manuscript. So, how does the ferroelectric ordering affect the interlayer hybridization. If there is no ferroelectric ordering in few-layer MoS₂, does the asymmetric response still exist?

2. Does the Si/SiO₂ substrate and h-BN contribute to the reflectance contrast spectroscopy? And do they affect the optical response of 3R-MoS₂?

3. The authors use a trilayer sample to construct the field-effect device, but use a pentalayer sample to demonstrate the asymmetric excitonic response. Why not use the same sample?

Reviewer #4 (Remarks to the Author):

The research by Swadep et al. provides valuable insights into the relationship between ferroelectricity and the excitonic response within parallel stacked transition metal dichalcogenide (TMD) monolayers. In their investigation, the authors focus on the as-grown five monolayers of MoS₂ in the 3R configuration and discover the presence of two distinct ferroelectric domains with Kelvin probe force microscopy (KPFM) measurements. These domains exhibited a significant difference of 127 mV. This discrepancy is believed to arise from potential layer sliding caused by shear strain during the exfoliation process. An intriguing aspect of their findings is the correlation observed in the KPFM maps and intensity map of $\Delta R/R$ within the 1906 to 1918 meV range. This direct relationship between ferroelectricity and excitonic response is a notable discovery in itself. Furthermore, the correlation is reinforced by the gate-dependent hysteretic reflectance contrast observed in the study. Additionally, Swadep et al. conducted a comprehensive analysis, which included transient reflectivity and transient ellipticity measurements. These additional experiments contribute to the robustness of the study's findings. A particularly noteworthy observation is the discrepancy in spin valley lifetime between the two ferroelectric domains. This observation holds significance, especially in the context of valleytronics in multilayer MoS₂. The research is further strengthened by theoretical calculations, which provide additional support for the observed phenomena.

Overall, the study conducted by Swadep et al. is unique in several aspects and holds potential for publication after revision listed below, which is necessary to further enhance the manuscript's quality.

(1) In page 2, first paragraph: "The appearance of different domains in a single crystal flake can be attributed to the sliding of constituting layers due to unintentional shear strain during exfoliation,¹⁹ as all the possible stacking configurations have similar energies (see SI. Fig.S10)." Moiré pattern-based different ferroelectric domains with sharp boundaries in artificially stacked TMD monolayers are well known, but observing different ferroelectric domains with sharp boundaries due to shear strain is not common. Therefore, further explanation is needed to understand the causes of local shear in that specific area (domain II). What can cause this local shear in that particular area?

(2) The inset of Figure 1d indicates that the separation between A and B excitonic features differs for different domains, suggesting tuned spin-orbit coupling in the two domains. Given the known differences in spin-orbit coupling between 3R and 2H MoS₂ (Nat. Commun. 11, 2391 (2020)), it is plausible that the two domains belong to different stacks. The authors are encouraged to comment on this possibility.

(3) The DFT-calculated absorption spectra appear to underestimate the band gap by 480 meV. It would be beneficial to present absorption spectra across the entire range, at least in the supporting information. Additionally, investigating the behavior of the C-exciton, which is generally observed due to band nesting, could provide further insights.

(4) The absence of optical and KPFM images from the device used in Figure 3 is notable. KPFM images are essential to observe different ferroelectric domains and compare them to gate-dependent maps. Including gate-dependent KPFM images of the devices would enhance the study's reliability. Moreover, providing device responses beyond the given voltage range (-0.2 to 0.2 V/nm) in the supplementary information would be beneficial. Moreover, for the sake of clarity it is advantageous to present the V_g not in units "V/nm" but just in "V". It particularly makes sense if the authors did not determine the gate oxide thickness for their certain wafer experimentally. Details of the field effect device fabrication and applying the gate voltage are not presented in "S1. Device Fabrication", which has to be improved.

(5) The author's claim regarding the utilization of the valley as a degree of freedom in multilayers (Page 8 last paragraph) needs discussion, considering that valleytronics in bulk TMDs (Nature 628, 746–751 (2024)) have been explored by other researchers as well. Incorporating a discussion on this topic in the introduction, supported by relevant literature, would provide context and depth to the study.

(6) Regarding the transient reflectivity and transient ellipticity measurements, it is unclear whether they were conducted at room temperature or low temperature (4K). Performing temperature-dependent measurements could be more informative, especially considering the noisy data observed in domain II, which could potentially be improved at low temperatures.

(7) Supporting information is not yet accurate and therefore requires a revision. Thus, e.g., in S1 one reference is missed. Minor point: Use the same style for capital and small letters in the captions of S1-6.

Reviewer #5 (Remarks to the Author):

Response to the reviewers:

We express our sincere gratitude to all the reviewers for their generous efforts in reviewing the manuscript and overall positive view of our work. At the same time, we are grateful for their critical assessments and the questions they posed. Their feedback has tremendously helped us underscoring the observations much more vividly and has, in our humble view, greatly improved the quality of our manuscript.

Following the referee remarks we have performed additional measurements, revised the manuscript accordingly. Below we fully address the points raised by the reviewers. The highlighted text refers to the changes made in the manuscript and the supplementary information.

We have also provided a 'List of Changes' in the main text at the end of our response.

Reviewer #1 (Remarks to the Author):

This article entitled "Excitonic signatures of ferroelectric order in parallel-stacked MoS₂" proposes optical diagnostics for interfacial ferroelectricity. The observations are supported by good-quality experiments and theoretical studies. The authors find the memory effect in a moiré-free 3R-MoS₂ crystal by the proposed method. The results of the manuscript are potentially interesting to related fields. Nevertheless, the main experiment needs to be described more carefully. The following points should be clarified or modified before publication.

We thank the referee for his/her positive assessment of our work and for raising the concern regarding the details in defining $\Delta R/R$. Indeed, in our original text, we failed to provide a satisfactory discussion on each of the quantities used in it. We have modified the main text to include these finer details and have also added additional data in the supplementary for completeness. Below, we provide a detailed response to their specific comments.

1. Authors should carefully describe how they calculated $\Delta R/R$. The intensity or reflectance spectroscopy for R_{ref} or R_{flake} could be included in the supplementary information.

Ans: We used $\frac{\Delta R}{R} = \frac{R_{Ref} - R_{Flake}}{R_{Flake}}$, where R_{Ref} is the reflection from Si/SiO₂/bottom-hBN/top-hBN (if top encapsulated) and R_{Flake} is the same from Si/SiO₂/hBN/MoS₂/hBN. We have added these details in the revised main text (page 2 last paragraph, page 6 last paragraph).

We have included the reflectance spectra R_{ref} and R_{flake} spectra in the supplementary information, (Fig. S3), with a reference to it in the main text (page 2 last paragraph).

2. What is the definition of R_{ref} ? Is it the position of hBN/SiO₂/Si without MoS₂ for Fig. 1c? What is the blue circle in Fig. 1c? Where is R_{ref} for the hBN-encapsulated MoS₂ in Fig. 3b?

Ans: In the older version of Fig. 1, we used the reflected spectrum from Si/SiO₂ as R_{ref}. For Fig. 3, we used Si/SiO₂/hBN/hBN as a reference point. To improve the consistency throughout the manuscript, we have now revised the spectra and/or intensity map in Fig. 1, Fig. S7, S8, S9, and S11(b) using Si/SiO₂/hBN as the reference point. We have added these details in the revised main text (page 2 last paragraph, page 6 last paragraph).

The blue circle in Fig. 1c marks an area on the seven-layer thick MoS₂, from where we extracted the blue-colored spectrum shown in Fig. 1d. We have added this clarification in revised Fig.1 caption.

3. The authors defined the contrast spectrum according to the definition in reference [20] and related it to the absorption coefficient. However, I wonder if this definition is correct. $\Delta R/R_{ref}$ seems to be correct according to past literature (Surface Science, 24 (1971) 417). If the definition is wrong, all results need to be recalculated.

Ans: We thank the reviewer for bringing up the significant subtlety in associating absorption spectra with $\Delta R/R$. It is evident from the literature that multiple approaches have been taken to calculate absorption-related spectra from reflectance spectroscopy. To accommodate different perspectives, we have recalculated all the results and replicated Fig. 1c-d and Fig. 3 with the redefined version, i.e., $\Delta R/R_{ref}$, and compared them side by side with the original version in SI Fig. S6 and S15. We have also done the same for an additional set of data, shown in Fig. S4 and S5. For a reference we have appended the results for Sample 1 (c.f. Fig. 1).

Figure R1: Reflectance contrast map and spectra of Sample 1 (from Fig. 1 of main text). We have used $\Delta R/R_{flake}$ on left panel and $\Delta R/R_{ref}$ on right.

It is empirically evident that both the definitions, $\Delta R/R_{ref}$ and $\Delta R/R_{flake}$ share similar characteristics, viz. the presence of momentum-indirect transition, X_A , and X_B features and their energetic positions. However, the choice of definition modifies the line shape of each transition and therefore, the overall appearance and the magnitude of the spectra. Notably, the key characteristics relevant for the current study, i.e., the separation and relative strength of the sub-features in X_A and X_B , largely remain unchanged. Therefore, the intensity maps yield similar results, and the conclusion of the work is not affected by the choice of specific approaches.

In the current main text, we choose to work with $\Delta R/R_{flake}$ due to the more prominent appearance of the momentum-indirect transition. However, to address this extremely serious issue, we have added a detailed discussion in the revised text (page 4 paragraph 1). We have also adjusted the text and refrained comparing the experimental data directly to absorbance of MoS₂ to avoid any possibility of confusion.

4. As the authors explain, $\Delta R/R$ is a physical quantity associated with the absorption, which is related to the thickness of the hBN film. Quantitative labels for the intensity of the reflection contrast should include the axis in Fig. 1d and Fig. 3, as in Fig. S5 or Ref [20] for the validity of experimental results.

Ans: We thank the reviewer for pointing it out. We have now included quantitative labels in the Y axis of revised Fig. 1d, 1d-inset, and Fig. 3

5. The energy is of XB at 2000meV in Fig. 3 smaller than in Fig. 1. What is the reason for the difference?

Ans: The relative red shift of $\sim 70\text{meV}$ of X_A , X_B in Fig. 3 with respect to Fig. 1 is due to the elevated sample temperature. The data shown in Fig. 3 were recorded at room temperature while the experiments for Fig. 1 were performed at 4K. Such a low-energy shift in X_A and X_B with increasing temperature has earlier been observed in monolayer MoS_2 as well [see, for example, S. Deb et al. Applied Surface Science 445, 542, 2018]. We have highlighted the nuance in the Fig. 3d caption.

Reviewer #2 (Remarks to the Author):

Due to the broken inversion symmetry resulting from a unique atomic registry, bilayer 2D materials such as hexagonal boron nitride and transition metal dichalcogenides (TMDs) can exhibit interfacial ferroelectricity, leading to rich physics within this fascinating system. While TMDs display distinct excitonic properties with wide-ranging optoelectronic applications, the impact of interfacial ferroelectricity on these properties remains largely unexplored. In this study, focusing on five-layer 3R-MoS₂, the authors identified two types of ferroelectric orders within a single MoS₂ flake and demonstrated the significant influence of these orders on intralayer excitonic transitions. Leveraging this finding, the oscillation strength of intralayer excitonic transitions was utilized as a non-invasive indicator to track the evolution of ferroelectricity orders under an external electric field. Additionally, the study revealed substantial differences in exciton population and spin-valley dynamics under the two ferroelectricity order conditions, with the authors proposing several explanations for these observations.

While the investigation topic holds promise, the manuscript currently lacks robust evidence to fully substantiate its arguments. Consequently, in my opinion it may not meet the standards required for publication in Nature Communications. Below are some questions for reference.

We thank the referee for his/her assessment that our research holds promise. Below, we address the questions raised in his/her report and further substantiate our arguments based on our new experimental results.

Q1: Fig. 1d, peak assignment of the Γ -Q transition might be suspicious. As claimed in the Abstract, “momentum-indirect excitons with out-of-plane electric dipole moment” refers to the Γ -Q exciton. Momentum-indirect transitions mostly manifest small oscillation strength in the optical reflectance/absorption spectrum since this physical process needs the assistance of phonons. Meanwhile, according to the manuscript, the optical measurement was conducted in a back-scattering geometry, while the Γ -Q exciton dipole moment is out-of-plane orientated, which translates to that the incidence parallels to the dipole moment, and this would further suppress the optical response of the Γ -Q transition in the reflectance spectrum. However, Fig.1d shows a much pronounced peak than the momentum-direct exciton transitions. For the author’s reference, the intense peak at lower energies were not observed in similar systems. (Nat. Nanotechnol. 2023, 18, 36–41; PRB. 2022, 106, 045411), and the authors would better provide more convincing evidences.

Ans: We thank the referee for this extremely insightful remark. We note that we tentatively assigned the low-energy transition to the Γ -Q exciton based on previous studies on similar systems (*ACS Applied Materials & Interfaces*, 13(48), 57588–57596, 2021; *ACS Photonics*, 10(4), 1159–1168, 2023) and the trend of red-shift with increasing layer number (*Nano Letters*, 10(4), 1271–1275, 2010). However, it is challenging to eliminate the possibility of the transition stemming from K-Q transition. **We have added this discussion in the main text (page 4 paragraph 2).** Nevertheless, this particular feature is related to the hybrid momentum-indirect excitons with out-of-plane electric dipole moment.

The transition was artificially enhanced in strength due to cavity effects in the specific sample featured in the main text. In order to provide a qualitative understanding, we have expanded our study by preparing additional samples. We now present a comparative analysis of room temperature $\Delta R/R$ data from four different samples, each prepared on (or encapsulated by) *h*BN of varying thicknesses. Our investigation accentuates the significance of thin-film interference in the Fabry-Pérot cavity formed

by Si/SiO₂/hBN/MoS₂/hBN (if top encapsulated), which dictates the apparent color of the flake and influences the intensity of different excitonic features at different energy ranges.

Figure R2: Room temperature reflectance contrast spectra of four different 3R-MoS₂ samples along with their optical microscopic images.

Samples with a turquoise appearance (left panels) shows a pronounced Γ -Q or K-Q and X_A peaks. Conversely, samples prepared on thicker hBN, appearing pink (central panels), exhibit no detectable $\Delta R/R$ intensity in the low energy range, and the X_A transition is only weakly visible. However, the X_B feature is relatively stronger in this sample. In the case of an orange-colored sample (top encapsulated, right panel), X_A and X_B transitions are prominently present, while the lower energy feature remains invisible. In a separate ongoing project, we are making an attempt to quantify such observations (which, according to our experiments, holds true for any TMD) alongside a complementary theoretical description.

We have added a description of the same in the main text (from page 2 last paragraph to page 4 paragraph 1) and have added the data in SI Fig. S4 - S5. We have also added these relevant works in our bibliography (Ref 18, 28)

Q2: The authors claimed that the two-domain structure was probably introduced by the shear strain during exfoliation, which causes sliding of constituent layers. In addition to the RC sub-features, the authors are encouraged to provide more evidence to exclude the moiré-ferroelectricity scenario.

Ans: We exploit the mechanical exfoliation using scotch tape to isolate few layer flakes from bulk 3R-MoS₂ crystals. While in theory, the exfoliation and dry stamping processes only involve a pulling force perpendicular to the surface, in practice a horizontal component of force is ubiquitously present. As these flakes remain anchored to the substrate, the horizontal forces generate shear strain, potentially triggering slip avalanches which eventually result in the formation of differently stacked domains.

To illustrate the mechanism, we fabricated an additional sample by introducing *intentional* shear (horizontal force) perturbation during the dry transfer of 3R-MoS₂ from PDMS to Si/SiO₂/hBN. The surface potential map obtained via KPFM (shown below) shows that this intentional horizontal force readily results in the formation of densely packed ferroelectric domains, resembling artificially stacked moiré interfacial ferroelectrics, in terms of the typical size of the domains. However, achieving precise control over domain formation and their structural organization remains an open problem. **We have added it to the revised main text (page 2 paragraph 3).**

Figure R3: (Left) Optical micrograph of a bi- and trilayer 3R-MoS₂ prepared on Si/SiO₂/hBN with deliberate shear perturbation (horizontal force). (Right) Surface potential map acquired by KPFM measurements.

The domain observed in the RC map in Fig. 1c has a direct corresponds with the surface potential map (Fig.1b) obtained from Kelvin probe force microscopy (KPFM). Notably, the KPFM image reveals no moiré periodicity on the sample. Hence, the conjunction of these two experimental results unequivocally eliminates the moiré effect as the underlying cause for the appearance of X_A sub-features. A similar domain structure has previously been observed in natural 3R-MoS₂ [S. Deb et al. Nature 612, 465, 2022]. It is crucial to acknowledge that the origin of ferroelectricity in both 3R-MoS₂ flakes and artificially prepared moiré transition metal dichalcogenide (TMD) where the triangular moiré domains atomically reconstructed to rhombohedral stacking, remains exactly the same. However, the typical moiré length scale is smaller or comparable to the diffraction-limited resolution of far-field spectroscopy. As a result, far-field probing in artificially stacked parallel stack would only result in a domain-averaged spectroscopic signature from the crystal. **We have expanded the preexisting discussion in the main text for further clarification (page 4 paragraph 2).**

Q3: Fig. 3e, there is an evident discontinuity in the junction of the forward and backward sweeping curves at $\sim \pm 0.18 \text{ V nm}^{-1}$, what could be the reason leading to this phenomenon?

Ans: In our reflectance contrast measurements, we use a quartz tungsten halogen lamp for excitation. It so happens that this type of excitation source suffers from an unintentional drift in intensity. Such unintentional variation in fluence resulted in discontinuities at the junction of the forward and backward sweeping. To address this issue, we conducted an analysis of the standard deviation (SD) which is proportional to FWHM (therefore, independent of intensity) of X_A and X_B spectral region from the same dataset.

$$SD = \sqrt{\frac{\sum_{x=x_{\min}}^{x_{\max}} \text{Intensity}(x) (x - x_0)^2}{\sum_{x=x_{\min}}^{x_{\max}} \text{Intensity}(x)}} - \frac{\sum_{x=x_{\min}}^{x_{\max}} \text{Intensity}(x) (x - x_0)}{\sum_{x=x_{\min}}^{x_{\max}} \text{Intensity}(x)}$$

where, $x_0 = (x_{\max} + x_{\min})/2$, x is the energy of interest

In the resultant plots (Fig. R4a, c), the aforementioned crossing artifacts have been mitigated. **We have now included this in the main text (page 7 paragraph 1) and the analysis in SI Fig. S17.** For reference we have appended the results here along with the intensity plots for comparison.

Figure R4: Integrated RC intensity and standard deviation profile at the A and B excitonic region as function of gate voltage. **(a)** Standard deviation with $x_{\max} = 1893.71 \text{ meV}$, $x_{\min} = 1793.71 \text{ meV}$. **(b)** Integrated intensity with $x_{\max} = 1854 \text{ meV}$, $x_{\min} = 1850 \text{ meV}$ **(c)** Standard deviation with $x_{\max} = 2045.11 \text{ meV}$, $x_{\min} = 1945.11 \text{ meV}$. **(d)** Integrated intensity with $x_{\max} = 2006.81 \text{ meV}$, $x_{\min} = 2002.81 \text{ meV}$.

Q4: Fig. 4d and 4e, the slow decaying of lower energy level signal in domain II was attributed to the slow state filling resulted from the interlayer charge transfer probably facilitated by the interlayer hybridization, however, comparing the amplitude of the TR ΔR signal of the lower and higher energy states, why the two domains exhibit an opposite behavior? Also, some experimental details are missing, such as pump fluence, instrument response function.

Ans: The TR ΔR signal, which corresponds to differential reflectivity, is expected to correlate with the steady-state $\Delta R/R$ spectra when absorption bleaching due to state filling occurs.

Our spectrally resolved measurements (Fig. 4b and c) clearly show that the TR ΔR signal does not show a sign change around the lower and higher resonance energies (shown by colored arrows). Instead of

the sign change, which would suggest a bandgap-renormalization-induced shift of the absorption features, we observe a unipolar rise in intensity with probe energy, indicating pump-induced bleaching. This observation enables us to link the observed intensity variation of TRΔR to the steady-state spectra, as previously mentioned.

Therefore, the opposite trend in the amplitude of the TRΔR signal at the lower and higher energy states from the two domains is a direct consequence of opposite relative strength of these sub-features in the steady-state spectroscopic signal. However, the microscopic processes related to a domain-dependent energy relaxation and state filling dynamics are beyond the scope of our present study. **We have now added this explanation in revised figure 4 caption.**

We regret not including the critical details about the experimental parameters, these have been added to the revised **methods part of the manuscript, which reads as: "... Probe and pump pulses were electronically time-synchronized with an auto- and cross-correlation width of ~300 and ~700 fs, respectively. The cross-correlation time of ~700 fs marks temporal resolution of our measurement. ... The pump beam was set to a power of 30μW and the probe beam was set to 20μW. Both the beams were focused with an 80X (NA=0.50, f=200) objective onto the sample. Therefore, the pump and probe fluence roughly correspond to a maximum of 1.7KW/cm² and 1.1 KW/cm², respectively. ..."** **We have also added a dedicated section on this topic in the supplementary information.**

Minor typos, Supporting Information, on page 1: "hot pickup method [cite] to transfer hBN flakes..."; "...a conductive coating acquired from XXX".

Ans: We sincerely apologize for the unintentional oversight and thank the reviewers for bringing it to our attention. **We have revised the supplementary information accordingly.**

Reviewer #3 (Remarks to the Author):

In the manuscript “Excitonic signatures of ferroelectric order in parallel-stacked MoS₂”, the authors use the reflectance contrast spectroscopy to reveal the ferroelectricity in few-layer 3R-MoS₂. They show that contrary to electrostatic perception approaches, the hyperspectral reflectivity imaging is able to discern the correspondence between ferroelectric stacking and intralayer excitons. They attribute the origin of this correspondence to the stacking-specific interlayer resonant hybridization by means of ab initio calculations. They realize the room temperature control and optical readout of multi-state polarization with hysteretic features in a field-effect device. They also reveal the spin-valley dynamics on timescales comparable to TMD monolayers. Overall, the manuscript is well prepared and may be valuable for multiple fields of ferroelectrics, optoelectronics, and valleytronics. I can recommend its publication if the following comments are addressed.

We thank the referee for his/her recommendation of our work for publication pending addressing his/her valuable comments.

1. The asymmetric response of the momentum-direct excitons at the K point is caused by the interlayer hybridization, but the essential relationship between ferroelectric ordering and interlayer hybridization is not clear. I understand that the ferroelectric ordering would lead to a built-in electric field which shift the conduction and valence bands, but it does not change the transition energy and thus has no contributions on the asymmetric response, as mentioned in the manuscript. So, how does the ferroelectric ordering affect the interlayer hybridization. If there is no ferroelectric ordering in few-layer MoS₂, does the asymmetric response still exist?

Ans: With this comment, the reviewer addresses a central point of our manuscript that we have tried to clarify in the revised version. At first glance, as mentioned in the comment above, the built-in electric fields due to ferroelectric order would act to just shift, both, valence and conduction bands by the same amount, thus leaving the transition energies between those bands unchanged. However, in a multilayer sample, we can consider the individual layers akin to individual quantum wells due to the layer localized character of the K states. The energy levels for some of them attain degenerate values with each other for partially polarized ferroelectric order, while they remain nondegenerate for the fully ordered stacking. The degenerate energy levels of different quantum wells resonantly hybridize and a consequent shift of energy levels occurs. This layer and band specific strength of interstate hybridization leads to unequal shifts of layer-projected conduction and valence bands. Thus, the stacking order-dependent hybridization is the core mechanism leading to the experimentally observed dependence of spectroscopic signatures on ferroelectricity. We have further this clarification in page 4 last paragraph, page 6 paragraph 2. We have also added the conduction band wavefunctions in the top most panels of revised i.e., Fig.2 (d-f)

To gain an insight on the spectral response in case of a non-polar MoS₂ we have conducted low-temperature reflectance measurements on 2H-MoS₂ of various thickness, as shown below. Evidently, none of the features, viz. the momentum-indirect transition, X_A, and X_B show any emergent multiplicity.

Figure R5: $\Delta R/R$ spectra from non-polar 2H-MoS₂ at low temperature.

2. Does the Si/SiO₂ substrate and h-BN contribute to the reflectance contrast spectroscopy? And do they affect the optical response of 3R-MoS₂?

Ans: hBN being a wide bandgap insulator does not result in any additional features in the reflectance spectrum in the visible energy range. However, thin-film interference in the Fabry-Pérot cavity formed by Si/SiO₂/hBN/MoS₂/hBN (if top encapsulated), which dictates the apparent color of the flake, significantly influences the intensity of different excitonic features at different energy ranges. In order to provide a qualitative insight, we have expanded our study by preparing additional samples. We now present a comparative analysis of room temperature $\Delta R/R$ data from four different samples, each prepared on (or encapsulated by) hBN of varying thicknesses. The major findings of this extended study are discussed in our response to Reviewer #2 above. We kindly request the reviewer to refer to Fig. R2 and subsequent discussion.

Given the seriousness of this topic we have added these data and discussion in the revised main text page 2 last paragraph – page 4 paragraph 1 and supplementary information Fig. S4 and S5.

3. The authors use a trilayer sample to construct the field-effect device, but use a pentalayer sample to demonstrate the asymmetric excitonic response. Why not use the same sample?

Ans: In this project, we were driven by two major motivations to conduct the field-effect dependent study on the 3R-MoS₂ sample. Firstly, we aimed to validate the assertion linking excitonic response with ferroelectricity, which should (as it has) manifest as a non-volatile or hysteretic response when the crystal is subjected to an external electric field. Secondly, we sought to explore the feasibility of domain switching in natural ferroelectric crystals, particularly beyond the bilayer (which has previously been reported in artificially prepared bilayer stacks by several research groups) - pivotal insights for scalability and potential applications.

However, a major experimental concern in a field-effect device is that the external electric field is only significantly present at the gate/insulator/sample interface and rapidly screened away from the

interface. This asymmetric gate field could introduce additional complexities, particularly in relatively thicker samples, potentially leading to erroneous interpretations of the results. Hence, to provide an unambiguous conclusion, we confined our study to trilayer systems.

In addition, we were also limited by a practical limitation to a certain extent. For the sample featured in the main text, Fig. 1 (and for most of our studies) we have used 285-300 nm oxidized Si substrate for their better optical contrast throughout the energy range of interest. The higher oxide thickness would drastically reduce the gate capacitance and the applied electric field for a given gate voltage. To mitigate this limitation, we fabricated the field effect devices on a 90 nm SiO₂/Si substrate in the current study (cf. Fig. 3), which allowed us to achieve the necessary electric field strength for our experiments.

Reviewer #4 (Remarks to the Author):

The research by Swadep et al. provides valuable insights into the relationship between ferroelectricity and the excitonic response within parallel stacked transition metal dichalcogenide (TMD) monolayers. In their investigation, the authors focus on the as-grown five monolayers of MoS₂ in the 3R configuration and discover the presence of two distinct ferroelectric domains with Kelvin probe force microscopy (KPFM) measurements. These domains exhibited a significant difference of 127 mV. This discrepancy is believed to arise from potential layer sliding caused by shear strain during the exfoliation process. An intriguing aspect of their findings is the correlation observed in the KPFM maps and intensity map of $\Delta R/R$ within the 1906 to 1918 meV range. This direct relationship between ferroelectricity and excitonic response is a notable discovery in itself. Furthermore, the correlation is reinforced by the gate-dependent hysteretic reflectance contrast observed in the study. Additionally, Swadep et al. conducted a comprehensive analysis, which included transient reflectivity and transient ellipticity measurements. These additional experiments contribute to the robustness of the study's findings. A particularly noteworthy observation is the discrepancy in spin valley lifetime between the two ferroelectric domains. This observation holds significance, especially in the context of valleytronics in multilayer MoS₂. The research is further strengthened by theoretical calculations, which provide additional support for the observed phenomena.

Overall, the study conducted by Swadep et al. is unique in several aspects and holds potential for publication after revision listed below, which is necessary to further enhance the manuscript's quality.

We thank the referee for his/her for a positive assessment and stressing the novelty of our work in a very compelling manner. We are also grateful for his/her recommendation of our work for publication pending addressing his/her valuable comments.

(1) In page 2, first paragraph: "The appearance of different domains in a single crystal flake can be attributed to the sliding of constituting layers due to unintentional shear strain during exfoliation,¹⁹ as all the possible stacking configurations have similar energies (see SI. Fig.S10)." Moiré pattern-based different ferroelectric domains with sharp boundaries in artificially stacked TMD monolayers are well known, but observing different ferroelectric domains with sharp boundaries due to shear strain is not common. Therefore, further explanation is needed to understand the causes of local shear in that specific area (domain II). What can cause this local shear in that particular area?

Ans: We thank the reviewer for raising this concern which is, in fact, exactly in line with the question raised by Reviewer#2. We kindly request the reviewer to refer to Fig. R3 and subsequent discussion.

Given the seriousness of this topic we have added the relevant data and discussion in the revised main text (page 2 paragraph 3) and supplementary information Fig. S2. We have also subdivided the main text in an additional section to emphasize the importance to the reader. Page 2 Section 'Ferroelectric domains in 3R single crystals'.

(2) The inset of Figure 1d indicates that the separation between A and B excitonic features differs for different domains, suggesting tuned spin-orbit coupling in the two domains. Given the known differences in spin-orbit coupling between 3R and 2H MoS₂ (Nat. Commun. 11, 2391 (2020)), it is plausible that the two domains belong to different stacks. The authors are encouraged to comment on this possibility.

Ans: To transition from a 3R stacking order to a 2H stacking configuration, one of the constituent layers must undergo a 60° rotation. However, in our typical sample preparation technique, such a rotation is improbable. Additionally, such a drastic rotation would typically result in large-scale fault lines, which have not been observed in the samples. To highlight the importance of stacking order, i.e., 3R and 2H on the spin-orbit coupling we have expanded our discussion in the introduction of the main text (page 2 paragraph 1 reference 23).

(3) The DFT-calculated absorption spectra appear to underestimate the band gap by 480 meV. It would be beneficial to present absorption spectra across the entire range, at least in the supporting information. Additionally, investigating the behavior of the C-exciton, which is generally observed due to band nesting, could provide further insights.

Ans: We note that DFT underestimates the band gap of semiconductors due to unaccounted derivative discontinuity of typical exchange-correlation functional as discussed in Phys. Rev. B 37, 10159 (1988). To achieve band gap energies that are more consistent with experimental data, we can apply a rigid energy shift estimated from GW calculations, as performed by Kim and Choi [PRB 103, 085404 (2021)]. For MoS₂, this estimate yields a band gap increase of about 530 meV. Applying this rigid shift to the band gap to the data in Fig 2m, we obtain a nearly quantitative agreement with experimental data, as shown below. We have elaborated on this correction in the main text of the revised manuscript (page 6 paragraph 3). We have also updated Fig. 2 accordingly.

Figure R6: Calculated absorption spectra in the A excitonic region based on intralayer dipole matrix elements and layer-specific exciton binding energy, including a phenomenological Lorentzian line width of 20~meV. The bottom X-axis incorporates a band gap shift of 530 meV

We appreciate and acknowledge the reviewer's suggestion to extend the spectral range of absorption spectra calculations. However, as noted in conjunction with the cavity effects provided by the *h*BN encapsulation (SI Fig. S4-S5), the broad, low-energy feature we associate with the indirect-gap transition is artificially enhanced by the particular choice *h*BN thickness used for the specific samples. Besides its dependence on the number of MoS₂ layers, we have not noticed any subtle features related to the ferroelectric order from it. From a theoretical perspective, calculating absorption features related to indirect transitions is significantly more challenging due to the involvement of phonons in

the absorption process, and would require substantially high computational efforts with little relevance to the main focus of the manuscript.

We anticipate a similar non-responsive behavior for C exciton. DFT calculations in the previous work by the first author [S. Deb et al., Nature 2022] show that the conduction- and valence-band states associated with the C exciton transition are only weakly polarized (and correspondingly layer-hybridized) for the investigated 3R bilayer. Hence, we can expect only weak effects of ferroelectric order on the C exciton transition energies.

Nevertheless, we believe that the reviewer's suggestions will provide a strong foundation for our future theoretical and experimental investigations aimed at exploring the intricacies of the band structure in ferroelectric TMDCs.

(4) The absence of optical and KPFM images from the device used in Figure 3 is notable. KPFM images are essential to observe different ferroelectric domains and compare them to gate-dependent maps. Including gate-dependent KPFM images of the devices would enhance the study's reliability. Moreover, providing device responses beyond the given voltage range (-0.2 to 0.2 V/nm) in the supplementary information would be beneficial. Moreover, for the sake of clarity it is advantages to present the V_g not in unites "V/nm" but just in "V". It particularly makes sense if the authors did not determine the gate oxide thickness for their certain wafer experimentally. Details of the field effect device fabrication and applying the gate voltage are not presented in "S1. Device Fabrication", which hast to be improved.

Ans: We regret not including the optical image and surface potential map of the device used for the experiments and results shown Fig. 3. We have now added the optical image and gate-dependent KPFM maps in SI Fig. S14 based on the reviewer's suggestions. We would like to point out that the KPFM measurements were performed almost a year after the initial sample preparation and optical measurements shown in the main text, so that some degradation of the sample due to repeated cooling cycles etc. is likely. We note that a substantially thick hBN (~85 nm) was chosen to encapsulate the sample from the top to ensure high reflectance contrast of X_A and X_B . However, this encapsulation presents significant challenges for performing KPFM measurements, as we have now mentioned in the revised text (page 6 paragraph 4). These challenges underscore the impact of our study, as the white-light reflectance measurements are not adversely affected by the hBN layer at all. After an extensive post-processing of the sample, including high-temperature annealing, contact mode ironing, and optimizing KPFM measurement parameters, we found that operating the AFM in tapping mode yields the best results for top-encapsulated samples. These experimental details have now been included in revised SI S2. Evidently, the KPFM images and the RC maps display similar characteristic features, such as the appearance of a ferroelectric domain roughly in the middle of the flake and its shrinking size with increasing gate voltage. However, we note that the exact appearance of the domain differs from the RC map, which we attribute to possible perturbation during the rigorous post-processing.

To further demonstrate the reliability of our technique, we have included data from an additional sample. This new data reestablishes a clear correspondence between the surface potential map obtained by KPFM and the broadband RC map. The additional data is presented in SI Fig. S10.

We have now used V_g in volt units instead of V/nm in the X or Y axes in all relevant plots (Fig. 3, S14, S15, S16, S17) and text. In SI Fig. S16, we have plotted $\Delta R/R$ as a function of gate voltage up to $\pm 20V$ (beyond $\pm 18V$ as in Fig. 3e). Since full switching of the ferroelectric order was achieved in this voltage

range, we did not explore higher values in order to avoid damaging the sample by dielectric breakdown.

We thank the reviewer for pointing out the missing details in the 'Device Fabrication' section. We have revised it and add the details of the field effect device fabrication and applying the gate voltage.

(5) The author's claim regarding the utilization of the valley as a degree of freedom in multilayers (Page 8 last paragraph) needs discussion, considering that valleytronics in bulk TMDs (Nature 628, 746–751 (2024)) have been explored by other researchers as well. Incorporating a discussion on this topic in the introduction, supported by relevant literature, would provide context and depth to the study.

Ans: We would like to thank the referee for pointing out this relevant recent work, which highlights the importance of TMD multilayers as potentially viable platform for valleytronics, in contrast to general notion based on early literature. In our humble opinion, our work would complement and add further appeal to the field of valleytronic with multilayer TMDs. However, we would like to bring out the subtle differences between these two studies.

As briefly discussed in the introduction of our manuscript, 3R stacking preserves the spin-valley locking otherwise associated with monolayer TMDs, even for bulk crystals, due to broken symmetries. In contrast, the 2H-polymorph lacks this trait and therefore, it is believed to be unattractive for valleytronic applications. The recent Nature work referenced above goes against this popular notion and utilizes tailored light fields to achieve valley polarization in a centrosymmetric bulk TMD, albeit on very short timescales below a picosecond. We have extended our introduction to include this topic and the potential to realize valley polarization also in 2H bulk TMDs with supporting literature (page 2 paragraph 1).

In our study, we exploit the intrinsic broken symmetry of 3R-stacking to show the generation and persistence of valley polarization over significantly longer time windows of about 10-15 ps. Depending on the specific domain structure, we can either achieve these long valley lifetimes, or rapid depolarization below the temporal resolution of our experiment, indicating the potential for electric control of valley dynamics simply through external gate voltages.

(6) Regarding the transient reflectivity and transient ellipticity measurements, it is unclear whether they were conducted at room temperature or low temperature (4K). Performing temperature-dependent measurements could be more informative, especially considering the noisy data observed in domain II, which could potentially be improved at low temperatures.

Ans: We apologize for the oversight in not mentioning a crucial experimental details in the manuscript and thank the reviewer for bringing it to our attention. We note that the transient reflectivity and ellipticity measurements were conducted at 4K. This detail has been included in Fig. 4 caption.

The noisy appearance of the transient ellipticity data on domain II suggests a vanishingly small (to instrument sensitivity) Kerr ellipticity signal beyond ~ 1 ps pump-probe delay. We would like to highlight that a similar noise floor is observed for the ellipticity signal on domain I beyond ~ 20 ps pump-probe delay, as depicted in the following figure. To make this observation apparent we have extended the X scale of Fig. 4f, g in the revised manuscript. We note that the Y scales in panels f and g (Fig. R7 c, d) are identical, so that ellipticity amplitudes and noise levels can be directly compared.

Figure R7: (a-b) False color map of transient ellipticity as a function of probe energy from domain I and II. (c-d) Transient reflection at selected energies, marked by colored arrows in the top panels.

To provide a comprehensive overview of the ellipticity data, we have added a false color map of transient Kerr ellipticity as a function of probe energy from domain I and II in SI Fig. S28.

(7) Supporting information is not yet accurate and therefore requires a revision. Thus, e.g., in S1 one reference is missed. Minor point: Use the same style for capital and small letters in the captions of S1-6.

Ans: We sincerely apologize for the unintentional oversight. We have revised the supplementary information accordingly and updated the section captions as per the reviewer's recommendation.

Reviewer #5 (Remarks to the Author):

Ans: We thank the reviewers for their generous efforts.

List of Changes:

Page 1:

1. Rephrased text “prevalent in a variety of”
2. Added text “of certain van der Waals materials”
3. Added text “along the out-of-plane direction”

Page 2:

4. Added text “This presents the prospect of using spin and valley as useful degrees of freedom in a robust multilayer platform. It is worth mentioning that recent experimental efforts also underscore the possibility of utilizing multilayer 2H-TMDs for valleytronic applications despite their centrosymmetric structure.” (in response to question raised by Reviewer 4)
5. Added section heading “**Ferroelectric domains in 3R single crystals**”. (in response to question raised by Reviewer 2 and 4)
6. Added subsequent text describing the process of deliberately inducing ferroelectric domains in 3R-MoS₂. (in response to question raised by Reviewer 2 and 4)
7. Added text “Here R_{Flake} is the reflection from MoS₂ and R_{Ref} is the same from hBN (SI Fig. S3)”. (in response to question raised by Reviewer 1)
8. Added text “Notably, the lowest energy feature despite being a momentum-indirect transition has a prominent appearance. This can be attributed to the favorable thin-film interference condition in the Fabry-Pérot cavity formed by Si/SiO₂/hBN/MoS₂ in the given energy range (SI Fig. S4 - S5 for additional examples and further discussion)”. (in response to question raised by Reviewer 2 and 3)

Page 3:

9. Revised figure 1. Added quantitative labels in the Y axis of panel d, d-inset. (in response to question raised by Reviewer 1)
10. Revised figure 1 caption. “(black - 5L domain I, red - 5L domain II, and blue - 7L)”. (in response to question raised by Reviewer 1)

Page 4:

11. Added text “We note that RC spectra, $\Delta R/R$ can also be calculated using $(R_{\text{Ref}} - R_{\text{Flake}})/R_{\text{Ref}}$. We found that both representations, $\Delta R/R_{\text{Ref}}$ and $\Delta R/R_{\text{Flake}}$ share similar characteristic viz. the presence of three distinct features and their energetic positions (SI Fig. S4, S5, S6, S15). However, the choice of definition modifies the line shape of each transition and therefore, the overall appearance of the spectra. Evidently, the lowest energy peak appears vividly in $\Delta R/R_{\text{Flake}}$ (SI Fig. S4 - S5, S10). Therefore, in the following we refer to $\Delta R/R_{\text{Flake}}$ as RC spectra unless mentioned otherwise.” (in response to question raised by Reviewer 1)
12. Added text “due to the red-shift in its spectral position with increasing thickness.”

13. Added text “to nanoscale reconstruction of moiré domains (as confirmed by the complementary KPFM measurements).” (in response to question raised by Reviewer 2)
14. Added text “Correspondence between RC maps and KPFM images has been observed in several samples of different thicknesses, see Fig. S10 for example.” (in response to question raised by Reviewer 4)
15. Added text “Based on the trend of red-shift with increasing layer number, as shown in Fig. S4 and previously studied on similar systems, one can tentatively assign the low-energy peak either to the Γ -Q or to K-Q hybrid-excitonic transition.” (in response to question raised by Reviewer 2)
16. Added text “and SI Fig. S12 - S13”
17. Added text “through hybridization”; “lowest energy conduction-band and”; “both at the valence and conduction band”. (in response to question raised by Reviewer 3)

Page 5:

18. Revised figure 2. Added panel d, e, f. (in response to question raised by Reviewer 3)
19. Revised figure 2 caption. “lowest-energy spin-up conduction band (d-f) and”. (in response to question raised by Reviewer 3)
20. Revised figure 2. Additional X-axis in panel m. (in response to question raised by Reviewer 4)

Page 6:

21. Added text “The mechanism described below is similar to the quantum correction that leads to level anti-crossing by lifting the energy degeneracy of coupled quantum systems.” (in response to question raised by Reviewer 3)
22. Added text “largely”; “This layer- and band-specific strength of interlayer hybridization”; “unequal”; “layer-projected”; “intralayer transition”. (in response to question raised by Reviewer 3)
23. Added text “due to the unaccounted derivative discontinuity of the typical exchange-correlation functionals ... By applying a rigid energy shift to the band gap, estimated from GW calculations, one can achieve transition energies that are more consistent with experimental data. For MoS₂, this estimate yields a band gap increase of about 530 meV, which nearly compensates the experimentally observed offset.” (in response to question raised by Reviewer 4)
24. Added text “which become a challenge for heterostructures”. (in response to question raised by Reviewer 4)
25. Added text “90 nm”. (in response to question raised by Reviewer 3)
26. Added text “therefore, R_{flake} is the reflection from $h\text{BN}/\text{MoS}_2/h\text{BN}$ heterostructure and R_{ref} is the same from $h\text{BN}/h\text{BN}$ ”. (in response to question raised by Reviewer 1)

Page 7:

27. Revised figure 3. Added quantitative labels in the Y axis of panel d, e. (in response to question raised by Reviewer 1)

28. Revised figure 3. X and Y scales has been changed from V/nm to V. (in response to question raised by Reviewer 4)
29. Revised figure 3 caption. "room temperature" (in response to question raised by Reviewer 1)
30. Revised text "-18V to 18V". in response to question raised by Reviewer 4)
31. Added text "and SI Fig. S16". (in response to question raised by Reviewer 4)
32. Added text "We also examine the standard deviation, which correlates with the full-width half maximum of X_A and X_B . Evidently, the resulting plots exhibit the same features (SI Fig. S17)". (in response to question raised by Reviewer 2)

Page 8:

33. Added text "(SI Fig. S28 for the complete spectrum)". (in response to question raised by Reviewer 4)
34. Added text "with a coercive field of roughly 0.03-0.035 V nm⁻¹. The relatively small switching field of ferroelectricity by domain wall sliding could facilitate efficient electrical control of various optical properties of 2D materials."

Page 9:

35. Updated figure 4. Extended X scale in panel f and g. (in response to question raised by Reviewer 4)
36. Revised figure 4 caption. "All the pump-probe measurements were done at 4 K". (in response to question raised by Reviewer 4)
37. Revised figure 4 caption. "The opposite trend in the amplitude of the TRΔR signal at the lower and higher energy states from the two domains is a direct consequence of opposite relative strength of these sub-features in the steady-state spectroscopic signal". (in response to question raised by Reviewer 2)

Page 10-11:

38. Added **Methods, Data Availability** declaration, **Author contributions**

REVIEWER COMMENTS

Reviewer #1 (Remarks to the Author):

Authors have adequately answered my questions and manuscript can be accepted for the publication.

Reviewer #2 (Remarks to the Author):

The authors have addressed my comments adequately. The paper can be published now.

Reviewer #3 (Remarks to the Author):

The authors have addressed all my comments, and edited their manuscript accordingly. I would like to recommend its publication.

Reviewer #4 (Remarks to the Author):

Most of the authors' answers are satisfactory, but a few conclusions remain unclear. The following questions need to be addressed before recommending the manuscript for acceptance.

- The answers provided raise further questions in deliberate shear perturbed samples. Although there is a contrast present in bilayer (2L) in the intentionally sheared 3R layers, in the trilayer (3L), there is an insignificant pattern or contrast within the range of approximately 100 meV. This raises doubts about the presence of domains in thicker layers just by accidental shear. Additionally, the sudden surface potential change in the five-layer (5L) 3R system in Figure 1 is around 300 meV and very sharp, which was not observed in the intentionally sheared system. Could the authors provide more clarity on this?
- In the optical image of the device shown in Figure S14a, there appears to be no top graphene contact for applying the gate voltage. Additionally, a sharp change in surface potential is observed below the 3R MoS₂ flake where there is no MoS₂. Could the authors comment on the existence of this particular domain below MoS₂, as it is not visible in the reflectance contrast in Figure 3a?
- Do the authors expect a similar trend in chirality-dependent low-temperature PL near resonance, as observed in Figures 4f and 4g, where polarization is significantly different in different domains? Performing such an experiment could provide more concrete evidence.

Addressing these questions will help clarify the authors' conclusions and enhance the manuscript's overall quality.

Reviewer #5 (Remarks to the Author):

I co-reviewed this manuscript with one of the reviewers who provided the listed reports. This

is part of the Nature Communications initiative to facilitate training in peer review and to provide appropriate recognition for Early Career Researchers who co-review manuscripts.

Response to the reviewers:

We express our sincere gratitude to all the reviewers for their affirmative evaluation of our manuscript.

Below we fully address the points raised by the reviewer#4. The highlighted text refers to the changes made in the manuscript and the supplementary information.

Reviewer #1 to #3 (Remarks to the Author):

We take this opportunity to thank all the reviewers for their sincere efforts of reviewing the manuscript. We also thank them for recommending the manuscript for publication in Nature Communications.

Reviewer #4 (Remarks to the Author):

Most of the authors' answers are satisfactory, but a few conclusions remain unclear. The following questions need to be addressed before recommending the manuscript for acceptance.

We are grateful for reviewer's recommendation of our work for acceptance pending addressing his/her comments.

- The answers provided raise further questions in deliberate shear perturbed samples. Although there is a contrast present in bilayer (2L) in the intentionally sheared 3R layers, in the trilayer (3L), there is an insignificant pattern or contrast within the range of approximately 100 meV. This raises doubts about the presence of domains in thicker layers just by accidental shear. Additionally, the sudden surface potential change in the five-layer (5L) 3R system in Figure 1 is around 300 meV and very sharp, which was not observed in the intentionally sheared system. Could the authors provide more clarity on this?

Ans: The process of slip avalanche gets triggered due to the imbalance between the inherent van der Waals interaction, which maintains the layers in a specific stacking order, and the external shear force. Similar observations have been made for both bi- and trilayer by Jing Lang, Zilang Ye, et al. in 'Shear Strain-Induced Two-Dimensional Slip Avalanches in Rhombohedral MoS₂' (Nano Lett. 2023, 23, 15, 7228–7235). Clearly, this process remains a viable mechanism for inducing ferroelectric domains regardless of the layer thickness. Consequently, we attribute the appearance of a ferroelectric domain in the five-layer sample (c.f. Fig. 1) and its absence in a trilayer flake (c.f. SI Fig. S2) to the stochastic nature of the triggering events.

Additionally, it is possible that the bulk 3R crystal also possesses domains due to stacking faults that are retained even after micromechanical exfoliation and stamping. **We have added this clarification in the revised main text.** However, such a structural characterization is beyond the scope of the manuscript.

We would like to point out that the surface potential contrast observed between domain 1 and 2 roughly equates to 127mV as mentioned in the main text and shown in SI Fig. S1. The color scale values of 1.70V and 2.00V in Fig. 1b reflect the full range (300 mV) of the false color map, however the surface potential change between domains is only a fraction of this range, namely 0.127V. In case of the bilayer sample (c.f. Fig. S2) we observe a similar sharp change in the surface potential values across domains and the average contrast between the ferroelectric domains (~0.103V) matches well with the value obtained for sample 1, as **we have shown in the revised SI Fig. S2.** The slight decrement can be

attributed to the doping, surface effects, etc. as reported earlier by the lead author in Nature **612**, 465, 2022 and in Adv. Mater. **2400750**, 2024.

- In the optical image of the device shown in Figure S14a, there appears to be no top graphene contact for applying the gate voltage. Additionally, a sharp change in surface potential is observed below the 3R MoS₂ flake where there is no MoS₂. Could the authors comment on the existence of this particular domain below MoS₂, as it is not visible in the reflectance contrast in Figure 3a?

Ans: We have now updated SI Fig. S14, demarcating the 3R flake and graphite and added a schematic of the device structure for clarity. Note that the few-layer graphene flake is not used as a top gate but contacts the bulkier extension of the 3R-MoS₂ crystal, which extends from the left edge of the trilayer region. Therefore, the graphite acts a source/drain contact for the field effect device.

The area outside the 3R flake consists of various insulating layers, viz. hBN and SiO₂. Therefore, V_{KP} on the area outside the flake is a measure of the contact potential difference between the KPFM tip and the hBN/hBN/SiO₂/Si, varying consistently with the external gate voltage. In Fig. S14c, the domain-like contrast below the MoS₂ flake can be attributed to unintentionally trapped charge puddles at various insulating interfaces of hBN/hBN and hBN/SiO₂. As a result, they exhibit a non-monotonic change with gate voltage.

- Do the authors expect a similar trend in chirality-dependent low-temperature PL near resonance, as observed in Figures 4f and 4g, where polarization is significantly different in different domains? Performing such an experiment could provide more concrete evidence.

Ans: The TRKE measurements reveals a spin relaxation time of 10-15 ps for domain I and a few picoseconds for domain II. However, in the few-layer 3R flakes, any photoluminescence emission at the A exciton energy is a hot luminescence, as the band structure has an indirect gap. We expect this luminescence to be emitted on the picosecond timescale before carrier relaxation into the band minima occurs and therefore, it does not properly capture the dynamics of spin relaxation. Thus, in a chirality-dependent low-temperature CW-PL measurements we expect only a shallow contrast in the degree of polarization from domain I to domain II. Without sufficient time resolution to resolve the PL dynamics, which occurs at the limits of the fastest available streak camera systems, it would be hard to interpret the resultant polarization values of CW-PL experiments.

Addressing these questions will help clarify the authors' conclusions and enhance the manuscript's overall quality.

Reviewer #5 (Remarks to the Author):

Ans: We thank the reviewers for their generous efforts.

List of Changes:

Main manuscript

Page 2:

1. Added text “either to inherent stacking faults in the bulk 3R crystal or”
2. Revised text “the latter occurs”

Supplementary

Page 5:

1. Modified Figure S2 to include surface potential profile line cut and modified caption accordingly.

Page 15:

1. Modified Figure S14 to include circuit of gate-dependent measurements, highlight outlines of 2D layers and add schematic lateral view of device. The caption was also modified accordingly.

REVIEWERS' COMMENTS

Reviewer #4 (Remarks to the Author):

The authors satisfactory addressed all risen questions. Therewith I recommend accepting the paper for publication.

Response to the reviewers:

We express our sincere gratitude to all the reviewers for their affirmative evaluation of our manuscript.

Reviewer #4 (Remarks to the Author):

The authors satisfactory addressed all risen questions. Therewith I recommend accepting the paper for publication.

We are grateful for the reviewer's recommendation for publication our work.